# Scoring Tumor-Infiltrating Lymphocytes in breast DCIS: A guideline-driven artificial intelligence approach

**Matteo Pozzi**                                          MPOZZI@FBK.EU

*Fondazione Bruno Kessler, Trento, Italy and University of Trento, CIBIO, Italy*

**Natalie Klubickova**                    NATALIE.KLUBICKOVA@RADBOUDUMC.NL

**Michela Campora**                            MICHELA.CAMPORA@APSS.TN.IT

**Frederique Meeuwsen**                  FREDERIQUE.MEEUWSEN@RADBOUDUMC.NL

**Joey Spronck**                              JOEY.SPRONCK@RADBOUDUMC.NL

**Carlijn Lems**                              CARLIJN.LEMS@RADBOUDUMC.NL

**Michelle Stegeman**                    MICHELLE.STEGEMAN@RADBOUDUMC.NL

**Leslie Tessier**                            LESLIE.TESSIER@RADBOUDUMC.NL

**Mattia Barbareschi**                      MATTIA.BARBARESCHI@APSS.TN.IT

**Jeroen van der Laak**                  JEROEN.VANDERLAAK@RADBOUDUMC.NL

**Giuseppe Jurman**                            GIUSEPPE.JURMAN@FBK.EU

**Francesco Ciompi**                      FRANCESCO.CIOMPI@RADBOUDUMC.NL

*Department of Pathology, Radboud University Medical Center, Nijmegen, The Netherlands*

## Abstract

This study focuses on the assessment of Tumor-Infiltrating Lymphocytes (TILs) in Breast ductal carcinoma in situ (DCIS) by integrating artificial intelligence with international guidelines. DCIS is a non-invasive cancer with intrinsic potential to evolve to invasive breast cancer (IBC), making it critical to understand factors influencing this progression. TILs are a prognostic biomarker in IBC, but their role in DCIS remains under-explored. This work proposes an automated pipeline for computing TILs scores using deep learning for DCIS segmentation and TILs detection, following the guidelines of the International Immuno-Oncology Biomarker Working Group. We report the inter-observer variability at TILs scoring among Pathologists and show that the AI-based TILs scores have good concordance with human assessments. Future research will aim to reduce false positives in DCIS segmentation and detection, support the reference standard with immunohistochemical staining, and expand the dataset to enhance the robustness of the TILs detection algorithm. Ultimately, this method aims to aid Pathologists in assessing the risk associated with DCIS lesions

**Keywords:** DCIS, ductal carcinoma in situ, TILs score computation

## 1 Introduction

Ductal Carcinoma In Situ (DCIS) is a non-invasive form of breast cancer that has garnered significant attention owing to its increasing incidence and potential to progress to invasive breast cancer (IBC). DCIS is characterized by the presence of neoplastic cells within the breast ducts, without invasion beyond the basement membrane. DCIS is both morpho-

logically and clinically heterogeneous precursor lesion which might be identified through mammographic breast cancer screening programs in some cases (Wiechmann and Kuerer, 2008). To improve patient outcomes and treatment strategies, it is crucial to understand the factors influencing the progression of DCIS to IBC. The neoplastic transition evolving to IBC involves a multistep process of progression through the stages of atypical hyperplasia, DCIS, and ultimately invasive carcinoma (Chen et al., 2020). Studies have shown that a significant proportion of high grade DCIS cases may progress to invasive breast cancer within 30 years (Groen et al., 2017).

The management of DCIS poses challenges with questions surrounding the necessity of surgery for low-grade DCIS and the role of active surveillance (Coleman, 2019). The standard of care for DCIS typically involves either breast conservative surgery or mastectomy for patients treated with conservative surgery, ASCO guidelines suggest a clearance of margins greater than 2 mm (Morrow et al., 2016) and possible omission of radiotherapy for selected low-risk women (Rakovitch et al., 2018). Recent study compared genomics profile from patients with an initial DCIS lesion and a later invasive recurrence, showing that in 18% of the cases the invasive recurrence was not genetically related to the DCIS, while in 75% of the cases a clonal relation between the two lesions was found, suggesting that tumor cells were not eliminated during the initial treatment of DCIS (Lips et al., 2022). In order to avoid overtreatment and to move toward personalized medicine, it is important to develop new biomarkers that help assessing the exact invasive potential of the DCIS lesion.

Tumor-Infiltrating Lymphocytes (TILs) are a well known prognostic biomarker in IBC, with high TILs density associated with better response to adjuvant or neoadjuvant therapy and with a positive outcome (Ibrahim et al., 2014; Denkert et al., 2015; Dieci et al., 2018). In the context of DCIS, the assessment of TILs is an emerging area of research, while the role of different TILs in DCIS is still not well defined (Dieci et al., 2018). Several studies have investigated the quantification of TILs in DCIS and their correlation with clinical outcomes (Komforti et al., 2020), their potential prognostic significant related to the underlying genomics instability (Toss et al., 2020), and their association with a second breast cancer event (Farolfi et al., 2020).

Automated analysis of the tumor micro environment and assessment of TILs through deep learning techniques is increasingly gaining interest in the immuno oncology community. In the field on DCIS researchers have focused on analyzing the spatial distribution of TILs between patients with pure DCIS and the ones with IBC adjacent to DCIS (Narayanan et al., 2021). Generative Adversarial Networks (GAN) have also been used to segment DCIS and the resulting segmentation mask used to compute a TILs score (Hagos et al., 2022).

In this study we have focused on identifying DCIS as well as other relevant morphological classes for an objective and reproducible TILs score computation according to the recommendations of the International Immuno-Oncology Biomarker Working Group on Breast Cancer. We collected and annotated hematoxylin and eosin (H&E) stained digital pathology Whole Slide Images (WSIs) from Santa Chiara Hospital and used the publicly available BRACS dataset as an external validation set (Brancati et al., 2022). Aim of the study is to provide Pathologist a tool to enable quantitative assesment of TILs abundance in DCIS.

## 2 Materials and Methods

### 2.1 Data collection

A total of 100 WSIs were used for this study. Data was obtained from two independent sources: Santa Chiara Trento Hospital, for a total of 55 WSIs used for training, and BRACS dataset, for the remaining 45 used for testing.

In our pipeline we did not use QuPath (Bankhead et al., 2017) annotations of DCIS areas available within the BRACS dataset. Pathologist annotated selected area of interests, using the same classes of the TIGER challange (Computational Pathology Group, 2022), but including only dcis in the in-situ tumor. The class included in the annotations where the following: tumor_associated_stroma, inflammed_stroma, healthy_glands, invasive_tumor, necrosis, dcis, rest.

Annotation on both dataset were made by trained Pathologist which sparsely annotated the slides using QuPath. All slides from both datasets were converted into tif format and saved at a resolution of 0.5 micron per pixel.

### 2.2 DCIS segmentation

In our study we used a version of nnUNet (Isensee et al., 2020) adapted for digital pathology applications (Spronck et al., 2023) for multi-class tissue segmentation in whole-slide images. Training parameters encompass a square patch size of 512 pixels, a mini-batch size of 18 samples, and input resolution of 0.5 micro per pixels. The training dataset was partitioned into a 5 fold cross validation. Training have been performed for 1000 epochs and model checkpoints have been saved based on the overall DICE score performance on the internal validation set. Regions in the WSI which were not annotated, have been excluded from loss calculation. Classes other than DCIS and rest where not used for the following part of the analysis and where therefore ignored. DICE score limited to the DCIS class have been computed on the external BRACS dataset. We limited our evaluation on this class since it was the relevant one for the TILs score computation. During training we adopted a weighted sampling strategy to increase the frequency of sampling of DCIS regions (see weights in Table 1). An example of a segmentation mask is reported in Figure 1

| Label | Weigth |
|---|---|
| rest | 0.125 |
| tumor_associated_stroma | 0.125 |
| inflammed_stroma | 0.125 |
| healthy_glands | 0.125 |
| dcis | 0.250 |
| invasive_tumor | 0.125 |
| necrosis | 0.125 |

Table 1: Annotation weights used during nnUNet training

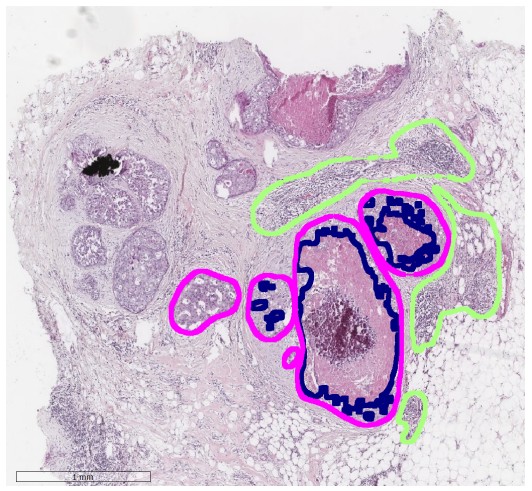

(a) Ground truth annotations

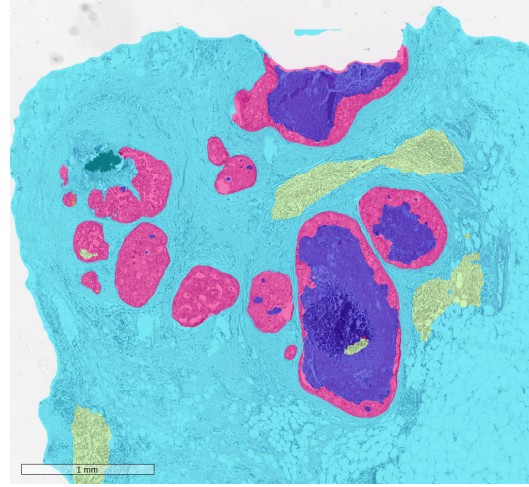

(b) Segmentation output from nnUNet

Figure 1: Example of the nnUNet segmentation output. DCIS is coloured in pink, necrosis in blue, inflammed stroma in yellow, while stroma in light blue. Images are visualized using the ASAP software.

## 2.3 TILs detection

TILs detection (see Figure 2) was done using Biototem algorithm and pipeline[1], the best performing algorithm in the computer vision task of the Tumor InfiltratinG lymphocytes in breast cancER (TIGER) challenge (Computational Pathology Group, 2022), hosted on the Grand Challenge Platform.

The model is based on UperNet with a Visual Attention Network (VAN) backbone, which enables long-range correlations in self-attention. The algorithm adapt the SFCN-OPI model, originally designed for nuclei detection, to improve TILs localization and reducing false positives.

We leveraged this model to identify TILs in BRACS slides. Detection results where then converted into xml format.

## 2.4 Pathologist TILs score

Two Pathologist provided TILs score for each WSI of the BRACS dataset following the International Immuno-Oncology Biomarker Working Group on Breast Cancer guidelines. Pathologist evaluated TILs score by assessing the stromal area around DCIS and reported a score per slide as a percentage of the stromal area infiltrated by lymphocytes. Pathologist scored the TILs ratio with 5% steps.

---

1. https://github.com/biototem/TIGER_challenge_2022

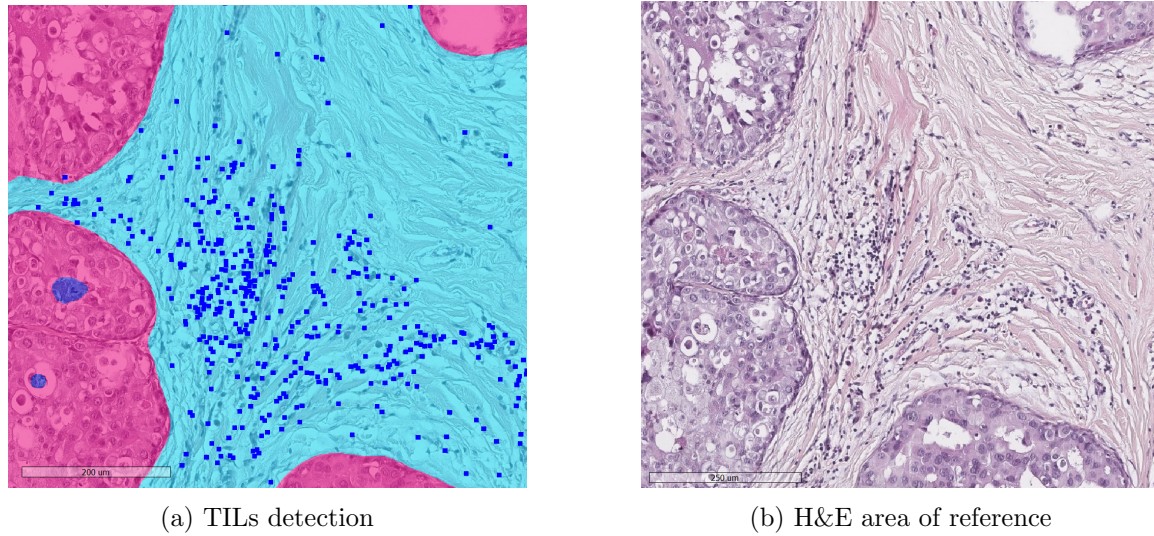

(a) TILs detection             (b) H&E area of reference

Figure 2: Biototem's TILs detection output

## 2.5 Automatic TILs score computation

DCIS segmentation mask derived from nnUNet model where converted into xml annotation files and merged with the TILs detection from Biototem algorithm. A buffer area of 250 micrometer where computed around each DCIS polygon annotation, merging buffers deriving from adjacent DCIS. The dimension of the buffer area is within the range suggested by the International guidelines, which suggest to inspect an area *up to* two high-power microscopic fields.

The buffer of a geometry is defined as the Minkowski sum of the geometry with a circle with radius equal to the absolute value of the buffer distance. A representative example is shown in Figure 3. We then counted the number of TILs detection within this buffer. We approximated the TILs area as being the one of a circle having a diameter of 8 micrometers (Swiderska-Chadaj et al., 2019). Tils score for each WSI is therefore computed as follows:

$$\text{TILs\_score} = \frac{\sum_{i=1}^{n} \text{TIL}_i \in (\text{B area - DCIS area}) \times 50 \ \mu\text{m}}{\text{B area} - \text{DCIS area}} \tag{1}$$

where B is the buffer derived from geometric operation. According to TILs scoring guidelines, immune hostspots identified by the class *inflammed_stroma* (i.e., regions with high-density clusters of lymphocytes within the stromal region) were included in the computation both by pathologist and by our computational approach.

## 2.6 Statistical Analysis

Performance of the DCIS segmentation model was evaluated using the DICE score. The correlation between the automatic TILs scores and the pathologists' scores was analyzed using Pearson and Spearman correlation coefficients to assess agreement.

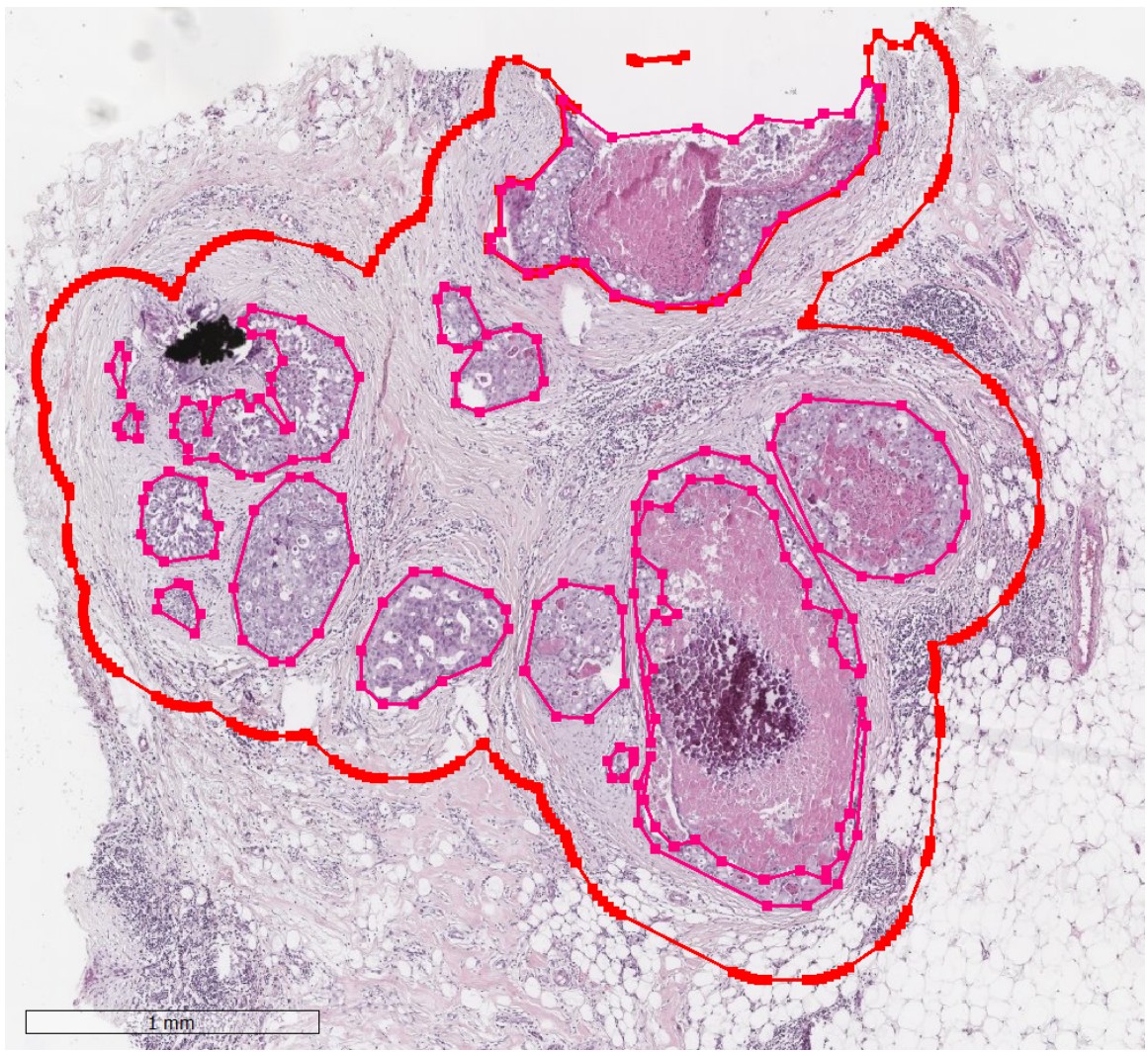

Figure 3: The buffer area used to computed tils detection. The total buffer area is represented in red, while the pink area represent the DCIS identified by nnUNet

## 3 Results and Discussion

The DCIS DICE score achieved by the model was of 0.69 with a variance of 0.08.

The computed TILs score was compared with the one estimated by Pathologist, resulting in a Pearson correlation coefficient of 0.53 with Pathologist 1 and 0.65 with Pathologist 2. Interestingly the correlation between Pathologist is limited to 0.48, showing a stronger concordance with the computed TILs score.

Spearman correlation coefficient showes a correlation of 0.57 between Pathologist 1 and 2. The computed TILs score shows a Spearman correlation of 0.61 with Pathologist 1 and 0.74 with Pathologist 2.

Interestingly, this result is in contrast with what was highlighted in the study by Hagos et al (Hagos et al., 2022) in which the concordance between Pathologist was higher than the concordance between Pathologist and AI-based computed score.

We developed a pipeline to automatically compute TILs scores following the guidelines for TILs assessment from the International Immuno-Oncology Biomarker Working Group. This study shows the discordance between Pathologists in computing density ratios with little to no quantitative guidance. On the contrary, computed TILs score is shown to correlate better with both Pathologist.

It has to be noted that the computed TILs score tend to be always low, potentially showing a tendency in underestimating the TILs score Figure 4. However this dataset did not contain any cases with high TILs score, therefore this hypothesis should be tested in future work, by integrating a wider and more diverse dataset.

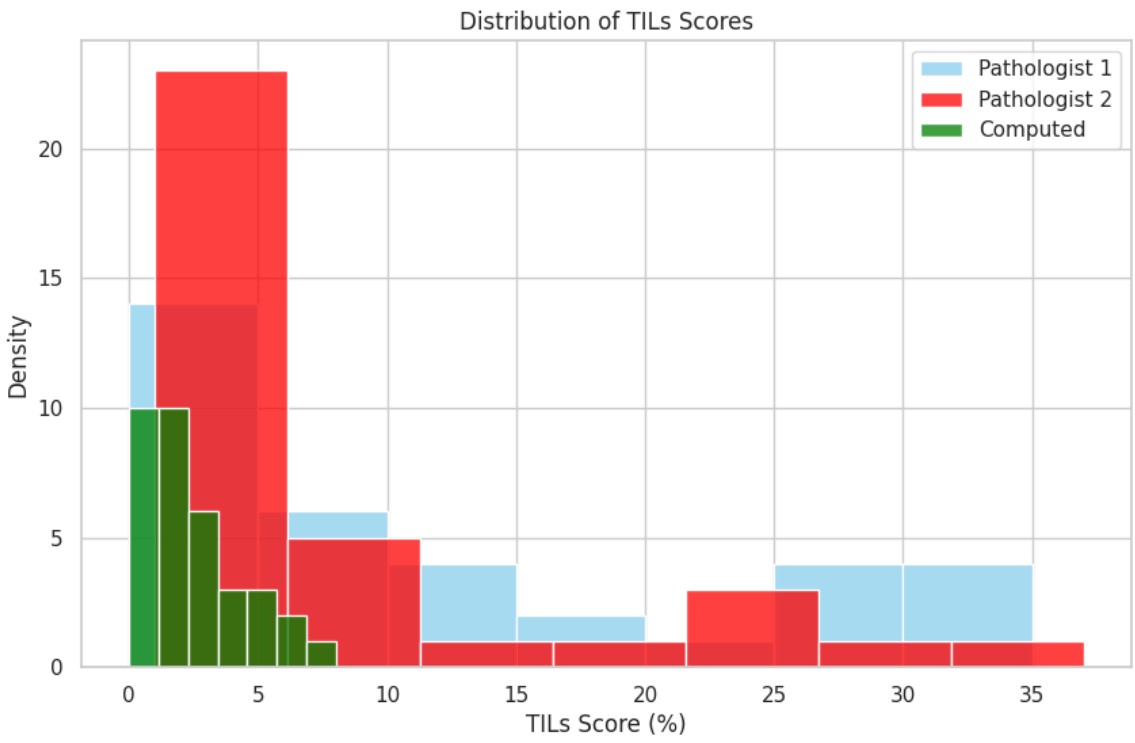

Figure 4: Distribution of TILs scores within Pathologist and the computed ones for the BRACS dataset

Future work will focus on reducing the number of false positives prediction in DCIS class as well as integrating immunohistochemistry staining to validate the predictions made by the TILs detection algorithm. Another point of action will be also increasing the size of the training dataset, with the goal to improve the segmentation performance of nnUNet. Additionally, in order to further investigate the robustness of these results, we plan to expand the number of pathologist who will assign TILs score for the BRACS validation dataset.

In conclusion we developed a pipeline to compute the TILs score in ductal carcinoma in situ using state of the art deep learning method for DCIS segmentation and TILs detection. Those output have been combined and exploited to compute TILs score following international guidelines. All results have been computed on an external and independent dataset, which has never been used in any part of the training protocol, to showcase the robustness of our method, with the aim to develop a tool able to assist Pathologist in DCIS risk assessment. Upon acceptance we aim to make our BRACS annotation publicly available, to faster the reproducibility of the study and to enlarge the number of high quality annotations, useful to develop large scale Deep Learning applications.

## 4 Acknowledgments

This work have been supported by the National Plan for Complementary Investments to the NRRP, project "D34H—Digital Driven Diagnostics, prognostics and therapeutics for sustainable Health care" (project code: PNC0000001), Spoke 2: "Multilayer platform to support the generation of the Patients' Digital Twin", CUP: B53C22006170001, funded by the Italian Ministry of University and Research.

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
