# OpenReview forum: "Scoring Tumor-Infiltrating Lymphocytes in breast DCIS: A guideline-driven artificial intelligence approach"
_MICCAI.org/2024/Workshop/COMPAYL — COMPAYL 2024_

### Official Review · Reviewer_Rvtz · 2024-07-08
**A potentially clinically relevant, fully supervised and pathologist-equivalent H&E scoring approach**

**Custom Rating:** 4
**Confidence:** 4

**Review:**

The authors provide a very clear application of a two step fully supervised pipeline to detect DCIS and TILs. Both components have been published previously and have been shown great performance. The TIL scoring system performance has been benchmarked against two pathologists, which leaves the question open on how it will work against consensus (majority) scoring employing 3 or more pathologists. While the system seems to outperform the pathologist scoring in terms of reproducibility, its clinical significance has not been shown. Further studies hopefully use treatment outcomes to assess this important and impactful aspect. Second, since the inter-rater agreement is relatively low (corr=48%), the problem may ask for additional ground truth, potentially using dedicated IHC "helper stains" (such as CD3, CD4, CD68, ..).

---

### Official Review · Reviewer_JqS6 · 2024-07-09
**Review of paper 23**

**Custom Rating:** 4
**Confidence:** 5

**Review:**

The article established a model based on nnUnet to identify DCIS regions for the purpose of combining it with the Biototem algorithm to quantify TILs metrics and compare the results with observations from pathologists.

The article's approach is based on clinical prior knowledge, making it quite comprehensible, but some details need clarification.

1. Are there specific references available for the guidelines of the International Immuno-Oncology Biomarker Working Group? If so, they should be provided.
2. The article claims to have collected and annotated multiple cohorts (end of Page 2) and used BRACS as an external validation set. However, it seems that all training data were actually sourced from Santa Chiara Trento Hospital.
3. What is referred to by "dcus" in Section 2.1?
4. When establishing the nnUNet, there is a category called invasive_tumor, but the article’s focus appears to be limited to DCIS. Does the use of slices involving invasive tumors conflict with the article's main topic?
5. The introduction of Biototem is too brief. From the git repo, it seems like a cell instance segmentation model. Please clarify its functionality.
6. If I understand correctly, the paper aims to measure the density of lymphocytes in the region of interest, but pathologists observe the area ratio. I have two questions: firstly, since the two metrics may be inconsistent but potentially monotonically related, would using Spearman correlation be more appropriate? Secondly, given that nnUnet includes Inflammatory_stroma, would directly calculating the area be a metric more aligned with pathologists' practices?
7. What is the y-axis in Figure 4? The number of slices? Based on the histogram counts, the sample size in Figure 4 seems to be approximately between 35 and 36. What is its source?

---

### Official Review · Reviewer_vT3T · 2024-07-12
**strong accept, a high quality, well written paper**

**Custom Rating:** 5
**Confidence:** 5

**Review:**

The submitted paper has high quality, has been written clear and well organized. The authors addressed a significant issue to assess the TILs as a potential biomarker in DCIS. The results demonstrate the success of the work by comparing the AI approach to pathologists.

Although The submitted paper is combining state of the arts like nnUnet in pathology for segmentation and the TIGER challenge winner method for detection as well as comparison of the AI and pathologists, there is slightly few novelty in pure AI parts. Also the results of the segmentation part expected to be higher or some other metrics can be used to represent the results better.

Please check the paper for typos, like “dcus” instead of “dcis” in section 2.1. or revising low informative part like table 1.

To conclude, I strongly recommend it for acceptance.

---

### Decision · Program_Chairs · 2024-07-16

Accept